# Functional Assessment of 12 Rare Allelic *CYP2C9* Variants Identified in a Population of 4773 Japanese Individuals

**DOI:** 10.3390/jpm11020094

**Published:** 2021-02-02

**Authors:** Masaki Kumondai, Akio Ito, Evelyn Marie Gutiérrez Rico, Eiji Hishinuma, Akiko Ueda, Sakae Saito, Tomoki Nakayoshi, Akifumi Oda, Shu Tadaka, Kengo Kinoshita, Masamitsu Maekawa, Nariyasu Mano, Noriyasu Hirasawa, Masahiro Hiratsuka

**Affiliations:** 1Laboratory of Pharmacotherapy of Life-Style Related Diseases, Graduate School of Pharmaceutical Sciences, Tohoku University, Sendai 980-8578, Japan; masaki.kumondai.s2@dc.tohoku.ac.jp (M.K.); aitoh0905@yahoo.co.jp (A.I.); gutierrez@tohoku.ac.jp (E.M.G.R.); noriyasu.hirasawa.c7@tohoku.ac.jp (N.H.); 2Advanced Research Center for Innovations in Next-Generation Medicine, Tohoku University, Sendai 980-8575, Japan; ehishi@ingem.oas.tohoku.ac.jp (E.H.); akiko.ueda.d1@tohoku.ac.jp (A.U.); 3Tohoku Medical Megabank Organization, Tohoku University, Sendai 980-8575, Japan; ssaito@megabank.tohoku.ac.jp (S.S.); tadaka@megabank.tohoku.ac.jp (S.T.); kengo@ecei.tohoku.ac.jp (K.K.); 4Faculty of Pharmacy, Meijo University, Nagoya 468-8503, Japan; 184331503@ccmailg.meijo-u.ac.jp (T.N.); oda@meijo-u.ac.jp (A.O.); 5Department of Pharmaceutical Sciences, Tohoku University Hospital, Sendai 980-8574, Japan; masamitsu.maekawa.a2@tohoku.ac.jp (M.M.); nariyasu.mano.c8@tohoku.ac.jp (N.M.); 6Laboratory of Clinical Pharmacy, Graduate School of Pharmaceutical Sciences, Tohoku University, Sendai 980-8574, Japan

**Keywords:** cytochrome P450 2C9, genetic variation, (*S*)-warfarin, tolbutamide, drug metabolism

## Abstract

Cytochrome P450 2C9 (CYP2C9) is an important drug-metabolizing enzyme that contributes to the metabolism of approximately 15% of clinically used drugs, including warfarin, which is known for its narrow therapeutic window. Interindividual differences in CYP2C9 enzymatic activity caused by *CYP2C9* genetic polymorphisms lead to inconsistent treatment responses in patients. Thus, in this study, we characterized the functional differences in CYP2C9 wild-type (CYP2C9.1), CYP2C9.2, CYP2C9.3, and 12 rare novel variants identified in 4773 Japanese individuals. These *CYP2C9* variants were heterologously expressed in 293FT cells, and the kinetic parameters (*K_m_*, *k_cat_*, *V_max_*, catalytic efficiency, and *CL_int_*) of (*S*)-warfarin 7-hydroxylation and tolbutamide 4-hydroxylation were estimated. From this analysis, almost all novel CYP2C9 variants showed significantly reduced or null enzymatic activity compared with that of the CYP2C9 wild-type. A strong correlation was found in catalytic efficiencies between (*S*)-warfarin 7-hydroxylation and tolbutamide 4-hydroxylation among all studied CYP2C9 variants. The causes of the observed perturbation in enzyme activity were evaluated by three-dimensional structural modeling. Our findings could clarify a part of discrepancies among genotype–phenotype associations based on the novel *CYP2C9* rare allelic variants and could, therefore, improve personalized medicine, including the selection of the appropriate warfarin dose.

## 1. Introduction

Genetic polymorphisms involved in drug absorption, distribution, metabolism, and excretion are important factors determining interindividual variations in drug responses observed when using identical dosing regimens [1,2,3]. The Clinical Pharmacogenetics Implementation Consortium (CPIC) provides a guideline to improve clinical pharmacology based on these genetic variabilities [4,5,6]. However, a detailed analysis that includes rare variant screening is needed for all major allelic variants because more than 80% of all variations in pharmacogenes have been observed at low frequencies in study populations [7].

Cytochrome P450 2C9 (CYP2C9) is a major drug-metabolizing enzyme that contributes to the metabolism of approximately 15% of clinically used drugs, including (*S*)-warfarin, a widely prescribed anticoagulant used as a long-term treatment for and preventive agent of thromboembolic events [8,9]. However, owing to its narrow therapeutic window, warfarin administration can cause heavy bleeding or thromboembolic events in some patients [10,11]. (*S*)-Warfarin, which is predominantly metabolized by CYP2C9, has a three- to five-fold anticoagulant effect when compared with that of (*R*)-warfarin; therefore, establishing the *CYP2C9* genotype of patients prior to drug administration could aid in predicting the outcomes of warfarin therapy [12,13]. According to several meta-analyses, patients carrying *CYP2C9*2* or **3* exhibit significantly lower metabolism of plasma warfarin than homozygotic *CYP2C9*1* carriers; hence, in these patients, lower maintenance doses are required [14,15]. To date, various warfarin-dosing algorithms, including those taking the *CYP2C9* genotype into account, have been established [10,16,17,18,19]. Washington University provides an online warfarin dose calculator that considers many different clinical factors and genotypes of *CYP2C9*, *CYP4F2*, vitamin K epoxide reductase, and gamma-glutamyl carboxylase genes (WarfarinDosing; http://www.warfarindosing.org accessed on 30 November 2020). Because most algorithms and available tools incorporate only four *CYP2C9* allelic variants (*CYP2C9*2*, *CYP2C9*3*, *CYP2C9*5*, and *CYP2C9*6*), unknown carriers of rare genetic variants are often misclassified as *CYP2C9* wild-type, placing these patients at higher risk of side effects and poor treatment outcomes, particularly considering that 2.1% of rare *CYP2C9* variants have been predicted to show deleterious functional effects [20].

The Pharmacogene Variation Consortium has defined more than 60 *CYP2C9* allelic variants (PharmVar; https://www.pharmvar.org/gene/CYP2C9 accessed on 31 May 2020), almost all of which were functionally characterized by assessing CYP2C9 enzymatic activity using recombinant CYP2C9 proteins [21,22,23,24,25,26,27,28,29]. Moreover, the correlation of *CYP2C9* genotype data in vitro and in vivo has been previously determined by characterizing 13 *CYP2C9* allelic variants identified in the Han Chinese population [30]. Hence, in vitro assays can help evaluate the impact of allelic variants in vivo without requiring highly invasive procedures or risking the induction of adverse drug reactions in clinical trial patients. In 2015, the Tohoku Medical Megabank Organization (ToMMo) reported the whole-genome sequences (WGSs) of 1070 Japanese individuals; these data were further expanded to include 4773 Japanese individuals in September 2019 (Tohoku Medical Megabank Organization; https://www.megabank.tohoku.ac.jp/english/timeline/20190913_01 accessed on 31 May 2020) [31]. Among the *CYP2C9* coding regions, 12 single nucleotide variations were newly identified from the WGS database. Characterization of the effects of the variants on enzymatic activity could provide support for improving clinical pharmacology.

Accordingly, in this study, we performed in vitro assays on CYP2C9.1 (wild-type), two representative variants (CYP2C9.2 and CYP2C9.3), and 12 novel structural variants identified in 4773 Japanese individuals. This assessment was performed using recombinant CYP2C9 protein expression in 293FT cells, with co-expression of cytochrome P450 oxidoreductase (CPR) and cytochrome b_5_. Additionally, we evaluated their substrate-dependent functional changes by assessing (S)-warfarin 7-hydroxylation and tolbutamide 4-hydroxylation. Furthermore, carbon monoxide (CO)-difference spectroscopy and three-dimensional (3D) structure analyses were conducted to gain further insights into CYP2C9 function.

## 2. Materials and Methods

### 2.1. Chemicals

Reagents were purchased from the following sources: (*S*)-warfarin, 7-ethoxycoumarin, and dimanganese decacarbonyl (Sigma-Aldrich, St. Louis, MO, USA); (*S*)-7-hydroxywarfarin (BD Gentest, Woburn, MA, USA); tolbutamide (FUJIFILM Wako Pure Chemical Corporation, Osaka, Japan); 4-hydroxytolbutamide and 4-hydroxytolbutamide-d9 (Toronto Research Chemicals, North York, ON, Canada); oxidized β-nicotinamide-adenine dinucleotide reduced form and β-nicotinamide-adenine dinucleotide phosphate reduced form (NADPH; Oriental Yeast, Tokyo, Japan); polyclonal anti-human CYP2C9 antibody (cat. no. ab4236; Abcam, Cambridge, UK); horseradish peroxidase (HRP)-conjugated goat anti-rabbit IgG (ProteinSimple, Tokyo, Japan); and sodium cyanide and cytochrome c from horse hearts (Nacalai Tesque, Kyoto, Japan). All other chemicals and reagents were of the highest commercially available quality.

### 2.2. Sanger Sequencing Analysis for the Detection of CYP2C9 Sequence Alterations

Peripheral blood leukocytes were isolated from whole blood of Japanese subjects participating in the community-based cohort study conducted by ToMMo; written informed consent was obtained from all subjects before sample collection [31]. Polymerase chain reaction (PCR) amplification was then performed using genomic DNA extracted from the cells with a Gentra Puregene Blood Kit (Qiagen, Hilden, Germany). The assay was carried out using more than 10 ng genomic DNA, 2× AmpliTaq Gold 360 Master Mix (Applied Biosystems, Foster City, CA, USA), and 0.5 µM each primer (Appendix A) in a total volume of 20 µL. The thermal cycling conditions included an initial denaturation at 95 °C for 10 min; followed by 30 cycles of denaturation at 95 °C for 30 s, annealing at 60 °C for 30 s, and extension at 72 °C for 1 min; and a final extension at 72 °C for 7 min. The PCR products were column purified and analyzed by Sanger sequencing using the same primers for each exon as in the PCR.

### 2.3. cDNA-Based Construction of Expression Vectors

Plasmids carrying CYP2C9*1 (wild-type) cDNA subcloned into the pENTR/D-TOPO vector were used as a template to generate 12 novel variant CYP2C9 constructs with a QuikChange Lightning Site-Directed Mutagenesis Kit (Agilent Technologies, Santa Clara, CA, USA) according to the manufacturer’s instructions. Plasmids carrying CYP2C9*2 or CYP2C9*3 cDNAs subcloned into the pENTR/D-TOPO vector were prepared as previously described [23]. All prepared wild-type and variant CYP2C9 cDNAs were confirmed by Sanger sequencing. The wild-type and variant CYP2C9 cDNA sequences were subsequently subcloned into the pcDNA3.4 mammalian expression vector (Thermo Fisher Scientific, Waltham, MA, USA). CPR, cytochrome b_5_, and mock cDNAs subcloned into the pcDNA3.4 vector were prepared as previously described [32,33].

### 2.4. Expression of CYP2C9 Variants in 293FT Cells

We cultured 293FT cells (Thermo Fisher Scientific) in Dulbecco’s modified Eagle’s medium (Nacalai Tesque) containing 10% fetal bovine serum at 37 °C under 5% CO_2_. Cells were plated at a density of 2.2 × 10^6^ cells/100-mm dish; 24 h after plating, cells were transfected with plasmids encoding CYP2C9, CPR, and cytochrome b_5_ cDNA (9.6 μg, 0.2 μg, and 0.2 μg, respectively) using 30 μL of 1.0 mg/mL Polyethylenimine “Max” (Polysciences, Inc., Warrington, PA, USA), according to previously described methods [32]. After incubation for 12 h at 37 °C, 0.25 mM 5-aminolevulinic acid hydrochloride (Nacalai Tesque) and 0.25 mM iron (II) sulfate heptahydrate (FUJIFILM Wako Pure Chemical Corporation) were added to the medium. After incubation for 48 h post-transfection at 37 °C, the cells were scraped from the plates. Microsomal fractions were prepared, and protein concentrations were determined as described previously [32].

### 2.5. Western Blotting

Immunoassays were performed using a Wes (ProteinSimple) and Compass for SW ver. 4.1.0 (ProteinSimple) (San Jose, CA, USA) to quantify CYP2C9 protein expression levels. Briefly, 100 ng of microsomes containing CYP2C9 wild-type and variant proteins were loaded into each well. CYP2C9 was detected using a polyclonal anti-CYP2C9 antibody (diluted 1:100) and HRP-conjugated goat anti-rabbit IgG. Following the manufacturer’s instructions, a total protein assay was performed to normalize each signal using 100 ng of microsomes. Recombinant human CYP2C9 supersomes (Corning Inc., Corning, NY, USA) were used as the standard to quantify CYP2C9 expression levels.

### 2.6. Determination of Cytochrome (CYP), Cytochrome P450 Oxidoreductase (CPR), and Cytochrome b_5_ Content

CYP2C9 holoprotein content, NADPH cytochrome c reduction activity, and cytochrome b_5_ content were spectrophotometrically measured according to previously reported methods using a Cary 300 UV-Vis (ultraviolet–visible) Spectrophotometer (Agilent Technologies) [32]. Data analysis was conducted using a Jasco Spectra Manager (JASCO Corporation, Sendai, Japan). Cuvettes (Sub-Micro Cells; 16.50-Q-10/Z20) were purchased from Starna Scientific, Ltd. (London, UK). CPR content determined by CPR activity was calculated based on previously reported methods [34].

### 2.7. (S)-Warfarin 7-Hydroxylation

(*S*)-Warfarin 7-hydroxylation by CYP2C9 was measured as previously reported, with several modifications [23]. The reaction mixture, in a total volume of 150 μL, consisted of the following components: the microsomal fraction (25 μg), (*S*)-warfarin (0.2, 0.5, 1, 2, 5, 10, 20, or 40 μM), and 100 mM potassium phosphate buffer (pH 7.4). Following pre-incubation at 37 °C for 3 min, reactions were initiated by the addition of 10 mM NADPH, with incubation at 37 °C for 60 min. Reactions were terminated by adding 150 μL of acetonitrile containing 25 nM 7-ethoxycoumarin as an internal standard. After protein removal by centrifugation at 15,400× *g* for 10 min, 10 μL of the supernatant was injected into an ultra-high performance liquid chromatography (UPLC)-fluorescence system consisting of an ACQUITY UPLC H-Class PLUS (Waters, Milford, MA, USA), ACQUITY UPLC FLR Detector (Waters), and an ACQUITY UPLC HSS C18 column (2.1 × 50 mm, 1.8-μm particle size; Waters) maintained at 40 °C. The mobile phase was a mixture of acetonitrile and water (40:60, *v*/*v*) containing 0.1% formic acid at a flow rate of 0.5 mL/min. (*S*)-7-Hydroxywarfarin content was measured at an excitation wavelength of 320 nm and an emission wavelength of 415 nm. Standard curves were constructed in the 12.5–6400 nM range using metabolite standards, with a quantification limit of 10 nM for (*S*)-7-hydroxywarfarin.

### 2.8. Tolbutamide 4-Hydroxylation

CYP2C9 tolbutamide 4-hydroxylation was measured as previously reported with several modifications [35]. The reaction mixture, in a total volume of 150 μL, consisted of the following components: the microsomal fraction (25 μg), tolbutamide (5, 10, 20, 40, 60, 80, 120, 200, or 400 μM), and 100 mM potassium phosphate buffer (pH 7.4). After pre-incubation at 37 °C for 3 min, reactions were initiated by the addition of 10 mM NADPH, with incubation at 37 °C for 20 min. Reactions were terminated by adding 150 μL acetonitrile containing 5 μM 4-hydroxytolbutamide-d9 as an internal standard. After protein removal by centrifugation at 15,400× *g* for 10 min, 5 μL of the supernatant was injected into a liquid chromatography-tandem mass spectrometry (LC-MS/MS) system.

4-Hydroxytolbutamide was measured using an LC-MS/MS system in the positive ion detection mode at the electrospray ionization interface (QTRAP 6500 LC-MS/MS system; AB Sciex, MA, USA). Separation by UPLC was conducted using a Nexera ultra-high performance liquid chromatography system (Shimadzu, Kyoto, Japan). Chromatographic separation was performed using a Kinetex C18 (2.1 × 50 mm, 5 µg particle size; Phenomenex, Shimadzu) maintained at 40 °C. Mobile phases were prepared using deionized water containing 0.1% formic acid as eluent A and acetonitrile containing 0.1% formic acid as eluent B. The flow rate was 300 µL/min, and the gradient program was as follows: initial elution with 10% B, followed by a linear gradient to 90% B from 3 to 6 min, and then immediately returned to the initial conditions and maintained for 3 min until the end of the run. Quantification analyses were performed in the selected reaction monitoring mode, in which ion transitions from the precursor into product ion were monitored, i.e., *m/z* 271.0 → 91.0 for tolbutamide (collision energy, 39 V), *m/z* 287.0 → 107.0 for 4-hydroxytolbutamide (collision energy, 23 V), and *m/z* 296.1 → 107.0 for 4-hydroxytolbutamide-d9 (collision energy, 25 V), using Analyst Software (Sciex). The optimized parameters for MS were as follows: entrance potential, 10.0 V; curtain gas, 25.0 psi; ion transfer voltage, 5000.0 V; temperature, 400.0 °C; gas 1, 50.0 psi; gas 2, 60.0 psi; and collision gas, 12.0. Standard curves for 4-hydroxytolbutamide were constructed in the 50–5000 nM range using metabolite standards, with a quantification limit of 50 nM.

### 2.9. Three-Dimensional Structural Modeling of CYP2C9

The 3D structural modeling of CYP2C9 was based on the CYP2C9 X-ray structure reported by Maekawa et al. (Protein Data Bank code 5XXI) [36]. (*S*)-Warfarin or tolbutamide was coordinated with the CYP2C9.1 model structure according to the CDOCKER protocol of Discovery Studio 2.5 (BIOVIA, San Diego, CA, USA). Docking iterations were conducted, considering the binding orientation and binding energies under the previously described conditions [33]. After replacing each substitution, structural optimization was performed according to previously reported protocols [37].

### 2.10. Data Analysis

Kinetic parameters (*K_m_*, *k_cat_*, *V_max_*, catalytic efficiency, and *CL_int_*) were obtained using the Enzyme Kinetics Module of SigmaPlot 12.5 (Systat Software, Inc., Chicago, IL, USA), a curve-fitting program based on nonlinear regression analysis. All values are expressed as means ± standard deviations (SDs) of experiments performed in triplicate. Statistical analyses for multiple comparisons were performed through variance analysis using the Dunnett’s T3 test or the Kruskal–Wallis method (IBM SPSS Statistics Ver. 22; International Business Machines, Armonk, NY, USA). The normality of our datasets was initially assessed using the Shapiro–Wilk test. Differences with *p* values less than 0.05 were considered statistically significance.

## 3. Results

Twelve novel structural variants with low allele frequency (0.01–0.05%) were identified by analysis of WGSs from 4773 Japanese individuals and confirmed by Sanger sequencing (Table 1). Additionally, already known variants encountered in this population were listed on the Appendix A.

The new variants were studied using 293FT cells for their heterologous overexpression together with CPR and cytochrome b_5_.CYP2C9 expression levels were normalized to total protein concentrations and quantified by western blotting with a polyclonal anti-CYP2C9 antibody that recognized all CYP2C9 variants (Figure 1 and Appendix A). CPR and cytochrome b_5_ content were confirmed to not differ significantly from those of the wild-type protein. The holoprotein content of CYP2C9 wild-type and variants expressed in 293FT cells were evaluated by CO-difference spectroscopy (Appendix A). Reduced CO-difference spectra of the microsomal fractions for wild-type CYP2C9 and eight variants (Cys13Arg, Arg125Cys, Lys138Met, Arg150Cys, Asn259His, Ser280Pro, Thr301Lys, and Met426Leu) were determined by measuring the increase in the maximum absorption at 450 nm after CO treatment. The maximum absorption at 450 nm was found to be low for five variants (Cys13Arg, Arg150Cys, Ser280Pro, Gly332Asp, and Gly417Ser. The remaining two variants (Glu400Ter (amino acid termination) and Arg433Trp) and two negative controls indicated no detectable holoprotein content. The characterization of all CYP2C9 variant microsomes is summarized in Table 2.

The kinetic parameters of CYP2C9 wild-type and variant proteins were determined by Michaelis–Menten curves for (*S*)-warfarin 7-hydroxylation and tolbutamide 4-hydroxylation (Figure 2 and Table 3 and Table 4). Kinetic parameters were calculated and normalized using total CYP2C9 protein content, quantified by western blot, using holoprotein and apoprotein containing microsomal fractions in order more closely reflect CYP2C9 enzymatic activity in vivo. The kinetic parameters from the CYP2C9 content determined by the CO spectra were presented in the Appendix A. The kinetic parameters of (*S*)-warfarin 7-hydroxylation for Thr301Lys, Glu400Ter, and Arg433Trp could not be determined because (*S*)-7-hydroxywarfarin and 4-hydroxytolbutamide were not sufficiently detected. The (*S*)-7-warfarin hydroxylation kinetic parameters for eight variants (CYP2C9.2, CYP2C9.3, Cys13Arg, Arg125Cys, Lys138Met, Ser280Pro, Gly332Asp, and Gly417Ser) were significantly lower than those of wild-type. Conversely, the *K_m_* values for CYP2C9.3, Arg150Cys, Asn259His, and Gly417Ser differed significantly from those of the wild-type protein. Moreover, CYP2C9.2, Cys13Arg, Arg125Cys, Lys138Met, Asn259His, Ser280Pro, Gly332Asp, and Gly417Ser had a significantly lower *V_max_* values, resulting in decreased *CL_int_* values. Furthermore, we evaluated the correlations between catalytic efficiency values of (*S*)-warfarin 7-hydroxylation and tolbutamide 4-hydroxylation and found a significant correlation between values among all CYP2C9 variants, as shown in Figure 3 (R^2^ = 0.919, *p* < 0.001).

To evaluate the molecular properties of CYP2C9 variants, we analyzed the 3D structural models of wild-type or variant proteins coordinated with (*S*)-warfarin (Figure 4). The Arg433Trp substitution variant formed a hydrogen bond with Arg97 and abolished the several interactions with Ser95, Ile112, Val113, Ser115, Asn116, Trp120, His368 and heme (Figure 5A). The (*S*)-warfarin docking analysis indicated that hydrogen bonds between Arg333 and Gln454 formation were changed by the Gly332Asp substitution, as shown in Figure 5B. In the Asn259His variant, electrostatic interaction and a hydrogen bond with Asp256 were formed alongside several interactions with His259 (Figure 5C). As a result, the hydrogen formation between Asn259 and Gln261 was abolished, and a hydrogen bond between Gln261 and His251 was formed. Additionally, Ser280Pro, located between the H- and I-helixes, lacked a hydrogen bond with Phe282, resulting in an abolished a hydrogen bond between Phe282 and Ile123 (Figure 5D). Additionally, there were no notable differences between both substrates coordinated with each CYP2C9 variant; and thus tolbutamide docking results are omitted.

## 4. Discussion

Within personalized medicine, the *CYP2C9* genotype plays an important role in improving the quality-of-life of patients because CYP2C9 catalyzes over 15% of clinically used drugs [9]. Moreover, patients have benefited from genotyped-based prescription and drug selection recommendations. To date, *CYP2C9*2*–**62* have been identified, and their variant proteins have been functionally characterized [21,22,23,24,25,26,27,28,29]. Although most alleles have been detected at low frequencies across populations, determining their functional effects is essential for appropriate drug treatment, particularly in unknown CYP2C9 allelic variant carriers, in whom there is a risk of genotypic misclassification, thereby increasing the risk of adverse reactions or treatment failure. Therefore, in the current study, we evaluated the impact of 12 *CYP2C9* variant alleles on enzymatic activity compared with that of CYP2C9.1–3.

Initially, two major variants (CYP2C9.2 and CYP2C9.3) were evaluated to determine whether the present system co-expressed with CPR and cytochrome b_5_ was suitable for the functional characterization of CYP2C9 proteins. Previous studies used mammalian cells to assess (S)-warfarin 7-hydroxylation and tolbutamide 4-hydroxylation, in which the extents of alteration of *CL_int_* values when compared to wild-type CYP2C9 were 25–47% (CYP2C9.2) and 8–28% (CYP2C9.3), respectively [27,28]. Although *CL_int_* values varied from studies due to expression methods, the present study demonstrated that the extent of alteration of *CL_int_* values for CYP2C9.2 and CYP2C9.3 was consistent with previous studies. Additionally, CYP2C9 and CPR ratios found in the microsomal fraction were comparable to levels found in in vivo studies [38,39]. Thus, the present methodology, CYP2C9 variants co-expressed with CPR and cytochrome b_5_, was used for variant functional characterization.

Western blotting results showed that only the Cys13Arg protein exhibited relatively low expression levels; because this substitution occurs in the membrane anchor region, its presence may alter topogenic functions. Almost all CYP proteins have hydrophobic segments in the N-terminus, which insert into the endoplasmic reticulum membrane [40,41]. The Cys13Arg substitution may influence membrane entrance by introducing a positively charged residue in the region, resulting in decreased CYP2C9 stability [42,43]. Further studies are required to elucidate the details of this phenomenon.

CO-difference spectroscopy is one of the most effective approaches for determining holoprotein structural stability [44]. Our current data confirmed that the absorption maximum at 450 nm tended to decrease alongside holoprotein stability [33,45,46,47,48]. Glu400Ter appeared to be unable to code the binding region and thus had no detectable holoprotein content or enzymatic activity. The Arg433Trp variant may have decreased holoprotein stability caused by absent interactions with Ile112, Val113, Ser115, Asn116, and Trp120 located in the substrate recognition site (SRS)-1, as well as differing interactions involving heme. Conversely, Gly332Asp and Gly417Ser variants, which had a relatively low absorption maximum at 450 nm, retained some enzymatic activity. Although the L-helix region of CYP enzymes contributed to the overall structural stability, this region is expected to be conserved to mediate CYP enzymatic activity; thus, the substitutions located around the L-helix region could affect holoprotein stability and decrease enzymatic activities [40]. Gly417, located in the F_1_-helix, is conserved in most CYP2 sequences; however, Gly417Asp did not appear to influence enzymatic activity, as was previously reported [40,49]. Our study detected further structural alterations in the well-defined PERF motif, which could decrease holoprotein stability.

The Asn259His variant located in the G and H-helices (G/H) loop showed substantially decreased *k_cat_* and *V_max_* values for both substrates. Alterations in the G/H loop may affect the electron-delivery system by ligand-induced conformational changes interacting with the C and D-helices loop [50]. Moreover, His259 influenced G/H loop flexibility, thereby decreasing holoprotein stability and enzymatic activity. Likewise, enzymatic activities were substantially decreased for the Ser280Pro variant, located around the end of the H-helix. The lack of hydrogen bonds around Pro280 expanded the H-helix, reducing I-helix flexibility. The I-helix, part of the SRS-4, is highly conserved within the CYP2 family; therefore, substitutions found in this region could influence enzymatic activity [40]. Importantly, hydrophobic amino acids located distal to the heme moiety of the I-helix correspond with the oxygen-binding pocket in CYP102, which may play critical roles in oxygen activation; hence the Ser280Pro substitution could decrease holoprotein stability and enzymatic activity.

Electron transport via CPR is essential for CYP enzymatic activity [51,52]. Positive charges in the C/D loop are primary CYP-CPR electron transfer sites [53]. The enzymatic activity of CYP2C9.14 and CYP2C9.35, which harbor the Arg125Leu substitution, has been reported to be slightly decreased [23,28]. Additionally, CYP2C9.8 (Arg150His) was associated with lower (*S*)-warfarin clearance caused by positive charge eliminations in the C/D loop [23]. Therefore, the enzymatic activities of Arg125Cys, Lys138Met, and Arg150Cys variants could decrease CPR electron transfer efficiency.

Recently, Arg125Cys was identified among 4773 Japanese individuals and was previously characterized as *CYP2C9*62* in Chinese individuals [29]. As previously reported, (*S*)-warfarin 7-hydroxylation and tolbutamide 4-hydroxylation were slightly decreased in these variants owing to their effects over heme or cytochrome b_5_ binding sites in CYP2 proteins. Notably, this variant has now been identified in two separate populations; therefore, other CYP2C9 variants characterized in this study may also be present in other Asian populations, a possibility that merits further investigation.

Interestingly, the Thr301Lys variant showed no enzymatic activity and but demonstrated a clear Soret peak at 450 nm. The threonine-containing region in the I-helix is a highly homologous sequence among mammalian microsomal cytochromes [40]. Several site-directed mutagenesis studies have stated the importance of Thr301 in forming the oxygen binding site and its influence over enzymatic activity [54,55]. Thus, the Thr301Lys substitution may influence electron transport to heme by decreasing oxygen binding capacity. In contrast, Lys301 caused higher steric hindrance than Thr301, thereby impeding substrate accessibility and decreasing affinity but not CO.

In this study, two representative CYP2C9 variants (CYP2C9.2 and CYP2C9.3) were characterized. Systematic reviews and meta-analysis have revealed that *CYP2C9*2* or *CYP2C9*3* carriers have decreased warfarin maintenance dose requirements [14,15]. Additionally, numerous studies have shown functional alterations in CYP2C9.2 and CYP2C9.3 in vitro. Therefore, CPIC provides guidelines for warfarin and phenytoin dosage based on the *CYP2C9* genotype [6]. Notably, five CYP2C9 variants (Lys138Met, Arg150Cys, Asn259His, Ser280Pro, and Gly417Ser) showed decreased enzymatic activity with comparable levels to that of CYP2C9.2; similar results were observed for CYP2C9.3 and three other CYP2C9 variants (Cys13Arg, Arg125Cys, and Gly332Asp), whereas Thr301Lys, Glu400Ter, and Arg433Trp showed no appreciable activity. Both catalytic efficiency and *CL_int_* ratios obtained for tolbutamide 4-hydroxylation were positively correlated with (*S*)-warfarin 7-hydroxylation, further supporting substrate specificity and highlighting the need for further assays with clinically relevant CYP2C9 substrates. In this way, our data supported that rare variant characterization could improve pharmacotherapeutic approaches. However, only representative variants are considered in most clinically used algorithms and predictive tools, warranting further investigation and inclusion of rare variant data in regional guidelines and clinical tools.

## 5. Conclusions

In this study, we characterized the enzymatic activities of CYP2C9.1–3 and 12 novel CYP2C9 variants expressed in 293FT cells. The majority of CYP2C9 variants described in this study showed altered enzymatic activity profiles. Consequently, carriers of these variants may have an increased risk of adverse effects or unfavorable treatment outcomes. However, further studies including additional biological replicates to discard the possibility of variations between replicates would be required because of using single microsomal preparation per CYP2C9 variant to reveal their qualitative features. Moreover, comparative studies to assess the difference between existing CYP2C9 crystal models and other protein data are required to clarify minor disparities. In conclusion, acquisition of rare variant data may help determine genotype-phenotype correlation trends, which could lead to the development of personalized medicine based on detailed information regarding *CYP2C9* genetic variations.

## Figures and Tables

**Figure 1 jpm-11-00094-f001:**
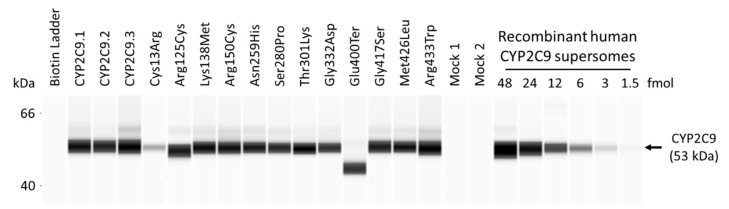
Representative western blots showing immunoreactive cytochrome P450 2C9 (CYP2C9) proteins. Average CYP2C9 levels were normalized according to total protein content. All assays and measurements were performed in triplicate using a single microsomal preparation. Recombinant human CYP2C9 supersomes were used to construct a standard curve through serial dilution (1.5–48 fmol) to quantify CYP2C9 total protein content. Ter represents amino acid termination. Mock 1 represents transfection with 10 μg mock plasmid. Mock 2 represents transfection with 9.6 μg mock plasmid, 0.2 μg cytochrome P450 oxidoreductase (CPR) plasmid, and 0.2 μg cytochrome b_5_ plasmid. N.D. represents not determined.

**Figure 2 jpm-11-00094-f002:**
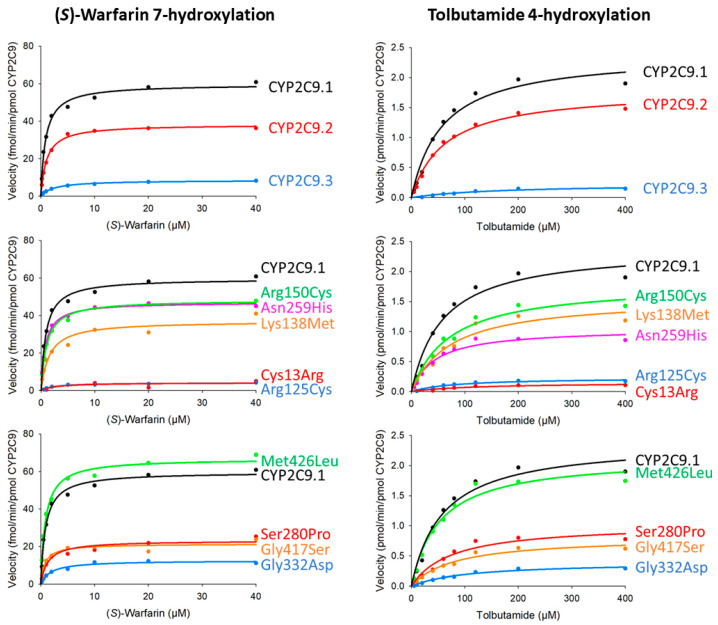
Michaelis-Menten curves for CYP2C9 variants. Determined kinetic parameters of (*S*)-warfarin 7-hydroxylation and tolbutamide 4-hydroxylation. All assays and measurements were performed in triplicate using a single microsomal preparation.

**Figure 3 jpm-11-00094-f003:**
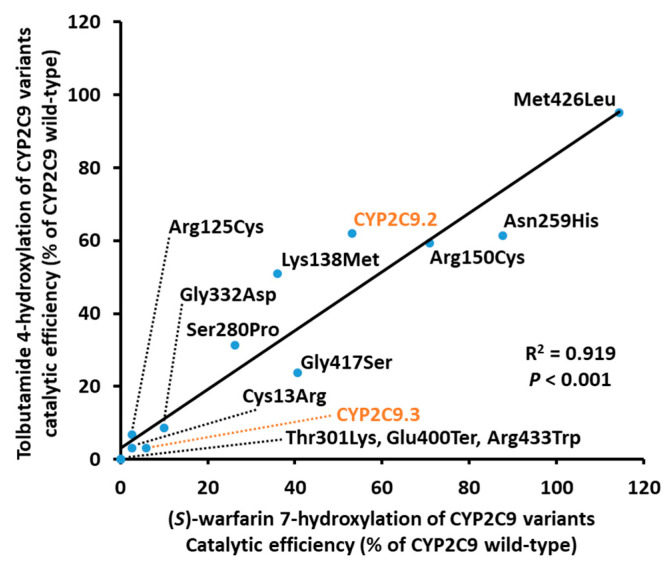
Correlation between the catalytic efficiency ratios for (*S*)-warfarin 7-hydroxylation and tolbutamide 4-hydroxylation among CYP2C9 variants. (*S*)-Warfarin 7-hydroxylation ratios are plotted on the horizontal axis, and tolbutamide 4-hydroxylation ratios are plotted on the vertical axis. Ter represents amino acid termination.

**Figure 4 jpm-11-00094-f004:**
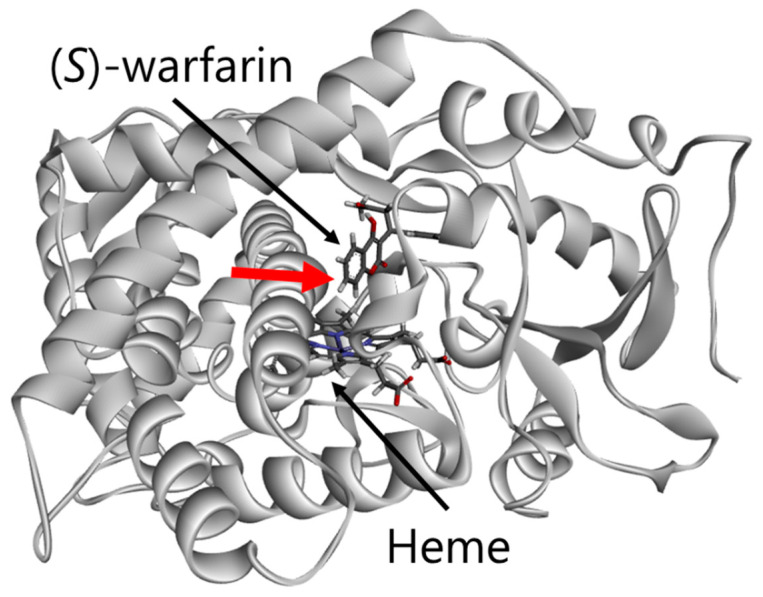
Diagram of the overall structure of CYP2C9. Red arrow represents metabolic site. The distance between central iron and metabolic site was 8.70 Å.

**Figure 5 jpm-11-00094-f005:**
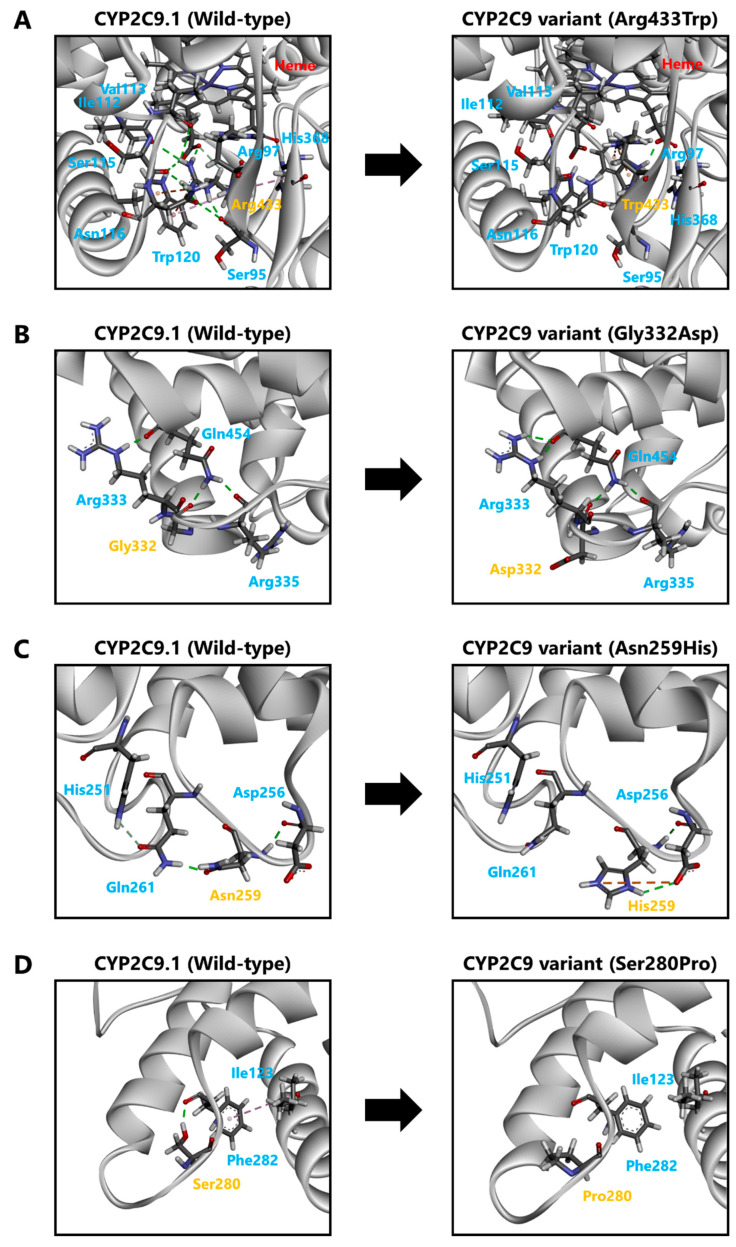
Diagram pairs showing the partial crystal structure of CYP2C9.1 (left image) and CYP2C9 variants (right image). Pink line: hydrophobic interactions. Green line: conventional hydrogen bonds. Gray line: carbon-hydrogen bonds. Orange line: electrostatic interaction.

**Table 1 jpm-11-00094-t001:** Novel *CYP2C9* allelic variants characterized in this study.

Nucleotide Mutations	rs Number	Amino Acid Substitutions	Frequency ^a^ (%)
37T > C		Cys13Arg	0.01
373C > T	rs375805362	Arg125Cys	0.01
413A > T		Lys138Met	0.03
448C > T	rs17847037	Arg150Cys	0.03
775A > C		Asn259His	0.02
838T > C	rs1312254054	Ser280Pro	0.02
902C > A		Thr301Lys	0.03
995G > A		Gly332Asp	0.02
1198G > T		Glu400Ter	0.01
1249G > A		Gly417Ser	0.01
1276A > C		Met426Leu	0.05
1297C > T	rs776908257	Arg433Trp	0.01

^a^ Allele frequency in 4773 Japanese individuals. Ter represents amino acid termination.

**Table 2 jpm-11-00094-t002:** The characterization of microsomes prepared from 293FT cells co-expressed with CYP2C9 variants, CPR, and cytochrome b_5_.

Variants	CYP Content (pmol/mg Protein)	CYP Holoenzyme Content (pmol/mg Protein)	CPR Content (pmol/mg Protein)	Cytochrome b_5_ Content (pmol/mg Protein)	CYP: CPR Ratio	CYP: Cytochrome b_5_ Ratio
CYP2C9.1 (wild-type)	207.25 ± 9.33	148.84 ± 6.41	214.14 ± 22.31	45.96 ± 3.59	0.97	4.51
CYP2C9.2 (Arg144Cys)	171.00 ± 6.08	149.45 ± 8.98	167.54 ± 17.23	39.66 ± 1.62	1.02	4.31
CYP2C9.3 (Ile359Leu)	252.74 ± 20.82	86.14 ± 6.97 **	205.88 ± 16.04	43.69 ± 5.27	1.23	5.78
Cys13Arg	51.56 ± 10.21	21.36 ± 1.61 ***	161.12 ± 11.40	17.91 ± 2.61	0.32	2.88
Arg125Cys	155.69 ± 5.00	71.49 ± 4.81 ***	242.71 ± 42.42	21.37 ± 2.91	0.64	7.28
Lys138Met	205.52 ± 11.64	160.29 ± 4.22	329.41 ± 33.03	31.43 ± 1.61	0.52	6.54
Arg150Cys	208.77 ± 9.39	24.87 ± 1.88 ***	410.39 ± 31.02	38.44 ± 4.24	0.51	5.43
Asn259His	176.58 ± 5.26	79.83 ± 8.53 *	276.70 ± 16.37	24.30 ± 3.01	0.64	7.27
Ser280Pro	149.27 ± 4.59	33.59 ± 0.76 **	227.28 ± 22.88	42.81 ± 3.29	0.66	3.49
Thr301Lys	191.66 ± 4.10	204.64 ± 7.20 *	370.58 ± 12.57	60.14 ± 4.25	0.52	3.19
Gly332Asp	151.22 ± 2.15	14.31 ± 0.60 **	277.25 ± 10.25	41.07 ± 1.77	0.55	3.68
Glu400Ter	159.68 ± 7.07	N.D.	328.46 ± 15.42	50.52 ± 4.58	0.49	3.16
Gly417Ser	189.31 ± 11.25	14.23 ± 2.07 ***	371.82 ± 36.04	51.20 ± 2.21	0.51	3.70
Met426Leu	181.17 ± 3.32	86.46 ± 3.90 **	354.21 ± 12.93	35.95 ± 2.86	0.51	5.04
Arg433Trp	188.41 ± 19.06	N.D.	453.54 ± 18.79	52.23 ± 2.70	0.42	3.61

Data represent the means ± SDs of the three independently performed catalytic assays. * *p* < 0.05, ** *p* < 0.01, and *** *p* < 0.005 compared with wild-type CYP2C9 by Dunnett’s T3 test. Ter represents amino acid termination. N.D. represents not determined. All assays and measurements were performed in triplicate using a single microsomal preparation.

**Table 3 jpm-11-00094-t003:** Kinetic parameters of (*S*)-warfarin 7-hydroxylation by microsomes from 293FT cells expressing wild-type and variant CYP2C9 proteins.

Variants	*K_m_* (μM)	*k_cat_* (fmol/Min/pmol CYP2C9)	*V_max_* (pmol/Min/mg Protein)	Catalytic Efficiency (*k_cat_*/*K_m_*) (% of Wild-Type)	*CL_int_* (µL/Min/Mg Protein) (% of Wild-Type)
CYP2C9.1 (wild-type)	0.88 ± 0.06	59.64 ± 0.86	12.36 ± 0.18	67.87 ± 4.30 (100.00)	14.07 ± 0.89 (100.00)
CYP2C9.2 (Arg144Cys)	1.06 ± 0.05	38.26 ± 2.49 *	6.54 ± 0.43 ***	36.00 ± 1.01 * (53.05)	6.16 ± 0.17 * (43.77)
CYP2C9.3 (Ile359Leu)	2.87 ± 2.11	8.74 ± 1.05 ***	2.21 ± 0.26 ***	3.98 ± 1.90 *** (5.87)	1.01 ± 0.48 *** (7.16)
Cys13Arg	2.36 ± 0.30	4.04 ± 0.35 ***	0.21 ± 0.02 ***	1.73 ± 0.19 * (2.54)	0.09 ± 0.01 ** (0.63)
Arg125Cys	2.45 ± 0.53	4.18 ± 0.19 ***	0.65 ± 0.03 ***	1.77 ± 0.47 ** (2.61)	0.28 ± 0.07 ** (1.96)
Lys138Met	1.52 ± 0.13	37.01 ± 0.98 ***	7.61 ± 0.20 ***	24.48 ± 2.16 ** (36.07)	5.03 ± 0.44 ** (35.76)
Arg150Cys	1.01 ± 0.10	48.23 ± 2.02 *	10.07 ± 0.42 *	48.16 ± 3.28 (70.97)	10.06 ± 0.69 (71.49)
Asn259His	0.79 ± 0.04	47.1 ± 2.02 *	8.32 ± 0.36 **	59.5 ± 2.41 (87.62)	10.5 ± 0.43 (74.66)
Ser280Pro	1.33 ± 0.24	23.15 ± 1.46 ***	3.46 ± 0.22 ***	17.79 ± 3.52 *** (26.21)	2.66 ± 0.53 *** (18.88)
Thr301Lys	N.D.	N.D.	N.D.	N.D.	N.D.
Gly332Asp	1.87 ± 0.27	12.50 ± 0.36 ***	1.89 ± 0.05 ***	6.75 ± 0.77 ** (9.94)	1.02 ± 0.12 * (7.25)
Glu400Ter	N.D.	N.D.	N.D.	N.D.	N.D.
Gly417Ser	0.78 ± 0.04	21.36 ± 1.01 ***	4.04 ± 0.19 ***	27.53 ± 1.39 * (40.56)	5.21 ± 0.26 * (37.05)
Met426Leu	0.86 ± 0.02	66.80 ± 1.55	12.10 ± 0.28	77.65 ± 3.15 (114.40)	14.1 ± 0.57 (100.01)
Arg433Trp	N.D.	N.D.	N.D.	N.D.	N.D.

Data represent the means ± SDs of the three independently performed catalytic assays. * *p* < 0.05, ** *p* < 0.01, and *** *p* < 0.005 compared with wild-type CYP2C9 by Dunnett’s T3 tests. Ter represents amino acid termination. N.D. represents not determined. All assays and measurements were performed in triplicate using a single microsomal preparation.

**Table 4 jpm-11-00094-t004:** Kinetic parameters of tolbutamide 4-hydroxylation by microsomes from 293FT cells expressing wild-type and variant CYP2C9 proteins.

Variants	*K_m_* (μM)	*k_cat_* (pmol/Min/pmol CYP2C9)	*V_max_* (nmol/Min/Mg protein)	Catalytic Efficiency (*k_cat_*/*K_m_*) (% of Wild-Type)	*CL_int_* (µL/Min/Mg Protein) (% of Wild-Type)
CYP2C9.1 (wild-type)	53.20 ± 1.86	2.29 ± 0.11	474.30 ± 28.95	43.04 ± 2.16 (100.00)	8.92 ± 0.55 (100.00)
CYP2C9.2 (Arg144Cys)	72.71 ± 6.84	1.93 ± 0.12	330.31 ± 24.89 *	26.65 ± 0.94 * (61.91)	4.56 ± 0.20 * (51.08)
CYP2C9.3 (Ile359Leu)	176.31 ± 7.07 **	0.24 ± 0.00 *	60.94 ± 0.97 *	1.37 ± 0.04 ** (3.18)	0.35 ± 0.01 ** (3.88)
Cys13Arg	104.64 ± 15.72	0.14 ± 0.01 **	7.14 ± 0.79 **	1.34 ± 0.08 ** (3.10)	0.07 ± 0.00 ** (0.77)
Arg125Cys	76.80 ± 4.17	0.22 ± 0.00 *	34.44 ± 0.82 *	2.89 ± 0.13 * (6.71)	0.45 ± 0.02 ** (5.03)
Lys138Met	68.98 ± 3.24	1.51 ± 0.05 *	310.31 ± 13.17 *	21.94 ± 1.27 * (50.97)	4.51 ± 0.32 * (50.53)
Arg150Cys	70.41 ± 2.38 *	1.80 ± 0.03	374.84 ± 8.34	25.55 ± 1.35 * (59.35)	5.33 ± 0.34 * (59.79)
Asn259His	36.90 ± 2.55 *	0.97 ± 0.03 *	171.40 ± 6.05 *	26.40 ± 1.47 * (61.33)	4.66 ± 0.32 * (52.26)
Ser280Pro	79.63 ± 10.54	1.06 ± 0.02 *	157.71 ± 4.22 *	13.47 ± 1.59 *** (31.30)	2.01 ± 0.29 *** (22.55)
Thr301Lys	N.D.	N.D.	N.D.	N.D.	N.D.
Gly332Asp	112.62 ± 32.21	0.40 ± 0.07 ***	61.17 ± 13.68 ***	3.71 ± 0.41 ** (8.61)	0.56 ± 0.08 ** (6.29)
Glu400Ter	N.D.	N.D.	N.D.	N.D.	N.D.
Gly417Ser	77.82 ± 3.07 *	0.79 ± 0.03 *	149.79 ± 6.81 *	10.20 ± 0.78 ** (23.69)	1.93 ± 0.18 ** (21.64)
Met426Leu	52.25 ± 2.99	2.14 ± 0.10	387.42 ± 22.18	41.00 ± 0.84 (95.16)	7.42 ± 0.19 (83.19)
Arg433Trp	N.D.	N.D.	N.D.	N.D.	N.D.

Data represent the means ± SDs of the three independently performed catalytic assays. * *p* < 0.05, ** *p* < 0.01, and *** *p* < 0.005 compared with wild-type CYP2C9 by Dunnett’s T3 test. Ter represents amino acid termination. N.D. represents not determined. All assays and measurements were performed in triplicate using a single microsomal preparation.

## Data Availability

Data available on request due to restrictions e.g., privacy or ethical.

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
