# Peer review of "Functional Assessment of 12 Rare Allelic CYP2C9 Variants Identified in a Population of 4773 Japanese Individuals"

_jpm, 2021, doi:10.3390/jpm11020094_

Round 1
Reviewer 1 Report
Review report
Manuscript jpm-1074537 by Kumondai et al, describes the functional characterization of 12 novel structural variants of CYP2C9 encountered in a Japanese population. This is of interest as CYP2C9 is an important CYP isoform in drug metabolism and genetic variability can be hold responsible of a larger part of encountered variability in drug efficacy and adverse drug reactions of specific types of drugs, such as warfarin. For this purpose, authors applied their recently described approach, using the 293FT cell model for heterologous over-expression of human CYPs together with CYP reductase CPR and cytochrome b5. Microsomal preparations were isolated for each of the CYP2C9 variants, and two specific CYP2C9 activities were measured. Based on activity outcomes relative to the CYP2C9 wildtype form, structural interpretations were made on the deviations caused by the mutations of the variants and their effects on CYP activity.
Although the objective is of interest, the described study suffers from several major shortcomings, namely missing validation of the new approach, errors in representation of data, inconsistent data, and missing of controls. These and some minor issues should be addressed.
Major issues
- The observed differences in enzymatic activity of the described variants can only be compared if the CPR:CYP and cytb5:CYP stoichiometry’s are similar among all tested variants. It’s known that relative ratios of CYP versus CPR and CYP vs. cyt b5 have a very strong influence on CYP activity (e.g. see PMID: 11095589). The manuscript describes a considerable variation in CYP2C9 protein expression (Fig 1) and a substantial variation in holoenzyme expression (Fig. 2). It is stated that CPR activity (i.e. cytochrome c reduction) and cyt. b5 content do not differ significantly among the studied variants (lines 229-230), however Supplementary Figures S2 and S3 demonstrate otherwise. This is even more relevant as several variants contain mutations which are suggested to interfere with CYP’s interaction with CPR and/or cyt b5 (lines 356-363). Still some variation of the two stoichiometry’s can be expected but similarities must be demonstrated, before correctly comparing activities among variants and drawing conclusion on the effect of the mutations. A Table should be presented in the main MS, describing the characterization of microsomes of all variants, mentioning the following items: CYP protein content (pmol/mg); CYP holoenzyme content (pmol/mg); CPR content (pmol/mg); cyt b5 content (pmol/mg); CPR:CYP ratio; cyt b5:CYP ratio. First four items should include their standard deviations. The CPR contents can easily be derived from the presented cytochrome c reduction activities, based on the molar turnover number of CPR in cytochrome c reduction (see: PMID: 19661994)
- From the manuscript, it’s not clear which CYP content of microsomes was applied for the determination of the enzymatic kinetic parameters. Are these holoenzymes or the CYP protein contents derived by immuno-detection? In this respect lines 225-226 are confounding.
- Recombinant human CYP2C9 (“supersomes”) was used as standard in the immune assays for quantification of CYP2C9 protein expression levels (lines 145-147). However, this essential control/standard is not included in the immune detection demonstrated in Figure 1.
- The applied approach using 293FT cells for the over expression of CYP2C9 together with its two redox-partners CPR and cyt b5 misses validation. Multiple studies have been reported on the enzymatic characterization of the CYP2C9.1, CYP2C9.2 and CYP2C9.3 variants with warfarin and/or tolbutamide. The current manuscript misses the discussion of results obtained with these three variants in comparison with former published data (including data from own laboratory), thereby validating the experimental approach. This discussion should be included in the first part of the Discussion section.
- Table 2 and 3 contain multiple errors and inconsistencies. Several of the calculated catalytic efficiencies, clearance values (CLint) and percentages contain errors; many of the values contain small deviations to actual values (e.g. Table 2, CYP2C9.3 Kcat/Km is 3.04 and not 3.98) but also bigger errors (e.g. Table 2, Arg125Cys percentage of CLint vs CYP2C9.1 is 18.8% and not 2%). Many other errors were detected in both Tables. Verify carefully and correct. For readability reasons, all values should be depicted with two decimal values, except for the percentage values which should be maintained with one decimal value.
Minor issues
- Change title to: “Functional assessment of 12 rare allelic CYP2C9 variants identified in a population of 4,773 Japanese individuals”.
- Lines 31-33: The cause of the observed kinetic differences was not only evaluated by the perturbation of the heme binding (holoenzyme content), as described in the Discussion section. Replace sentence by: “The causes of the observed perturbation in enzyme activity were evaluated by three-dimensional structural modeling.”
- Lines 33-34: no study was performed with individuals’ carriers of the rare variants, and as such discrepancies between genototype-phenotype are not clarified. However, findings “could clarify” such discrepancies. Adjust accordingly.
- Line 84: change to “ ……, and 12 novel structural variants identified….”
- Lines 86-88: awkward sentence, replace with: “This assessment was performed using recombinant CYP2C9 protein expression in 293FT cells, with coexpression of cytochrome P450 oxidoreductase (CPR) and cytochrome b5.”
- Lines 142 and 145: one loads quantities not concentrations: this was 25 micrograms of microsomes?
- Line 220: “Twelve novel structural variants with low….”
- Lines 220-224: it would be highly interesting to know which other, already known variants (CYP2C9*1-*60) were encountered in this population. This could be added in Table format to the Supplementary material file.
- Lines 224-225: An introductory sentence for the next section seem to be missing, such as “The new variants were studied using 293FT cells for their heterologous overexpression together with CPR and cyt b5.”
- Lines 227-228: confusing sentence: the Cys13Arg variant showed lower expression but no significant difference with the wildtype variant?
- Lines 228-230: see Major issues point 1); adapt this sentence;
- Lines 237-245: confusing, CO difference spectra of all studied variants are depicted in Figure 2. For which variants the holoenzyme content could be determined? See Major Issue point 1);
- Page 7: consider placing Figure 2 in Supplementary document;
- Page 9, Figure 4: Correlation plots A and B are virtually the same: catalytic efficiency (Kcat/KM) and CLint (Vmax/Km) are directly correlated; delete Figure 4B or place in Supplementary document;
- Page 11, Figure 5 is of high interest, but diagrams are very poor due to the use of a black background; improve by increasing resolution and the use of a white background
- Lines 323-324: it is an already a long-known phenomenon that the absorption maximum at 450 nm in CO difference spectra tends to decrease with increase holoenzyme instability of mammalian CYP proteins. Incorrect citations, adjust accordingly;
- Lines 329-330: Variants Gly332Asp and GLY417Ser did not retain their activity (9.9 and 40.6 % resp. for warfarin 7 hydroxylation, and 8.6 and 23.7% resp for tolbutamide 4 hydroxylation) and thus retained “some” activity; adjust accordingly
- Line 352: Reference 49 is not very appropriate; use Hlavica 2015 (PMID 26002739);
- Lines 365-366: “the Thr301Lys variant showed a slight decrease in enzyme activity”, but Table 2 and 3 depicted no activity at all for this variant. Correction necessary;
Reviewer 2 Report
The manuscript titled "Functional assessment of 12 allelic CYP2C9 variants identified in 4,773 Japanese individuals" reports characterization of these novel variants using (s)-warfarin and tolbutamide as substrates.
However, there are some major concerns that include:
1) The manuscript reports analysis of 12 novel variants with an allele frequency of 0.01 to 0.05 % in Japanese individuals. It is unclear what these variations represent and if these could be considered as polymorphisms or just mutations in individuals? These are novel but it is not clear whether how common are these with such a low frequencies. Can these even be considered as polymorphisms since these are under 1% of studied population?
It is stated that the maximum absorption at 450 nm was low for Gly332Asp and Gly417Ser, and that of the remaining two variants (Glu400Ter and Arg433Trp). However, from the figure 2 it is also very low for Cys13Arg, Arg150Cys, and Ser280Pro. Overall, 7 out of 12 studied novel variants have significantly low P450 content.
2) The 3D modeling was based on the crystal structure of CYP2C9 or 1OG5 bound to warfarin. However, in the 1OG5 structure, the warfarin is not bound or coordinated near heme. It is located in the access channel that is further away from the active site heme. Importantly, the 1OG5 structure, solved almost 20 years ago had several artificial mutations incorporated in the protein for purification and crystallization purposes. It is not clear why the recently available structures of CYP2C9 - Protein Data Bank code 5XXI for wild-type and 5X24 for CYP2C9*3 were not used. Although these were bound to losartan, but the protein does not include those artificial mutations incorporated in 1OG5 prior to purification. Additional 3D modeling studies using 5XXI and 5X24 protein data bank code may be needed to compare for differences to the current studies using 1OG5.
It is not clear how warfarin docked since no diagram showing warfarin docking is shown or mentioned. Did it dock in the active site or was it consistent to the crystal structure of 1OG5 with warfarin located in the access channel?
Were the 3D modeling studies done using tolbutamide to determine if the results that include molecular properties and hydrogen bonding were consistent to that observed with warfarin docking? Not sure why tolbutamide was omitted in the 3D modeling studies.
Overall, I believe further explanation of concern (1) above will be needed. Importantly, 3D modeling showing docked warfarin in the current and comparison using 5XXI protein data bank code will be necessary to validate the results. Lastly, similar docking studies with tolbutamide would be required to compare the results and conclusion from warfarin.
Reviewer 3 Report
This is an interesting study that has been carried out carefully.
Although there are two X-ray structures with better resolution than 1OG5.pdb
available (1R9O and 5W0C) the overall structure is not affected and thus has
no implications on the results, here.
Minor issues:
(1) Use "substantial", "considerable", or "notable" instead of "significant"
throughout the text, unless statistical p-values are available to underline
the corresponding statements.
(2) The observed high correlation of (S)-warfarin to the tolbutamin hydroxylation in the mutants is worth being noted in the abstract.
(3) Figure 5B shows the vicinity of the mutated residue 332, but not the
binding position of the docked warfarin and possible implications.
Here, docking of (S)-warfarin should reveal following points:
(a) In the wild-typ enzyme docking should put (S)-warfarin in a pose
from which 7-hydroxylation is possible (close to the HEM-iron).
Add an additional Figure for this.
(b) In the mutations that show (drastically) decreased warfarin hydroxylation
the corresponding docking pose of warfarin is either expected to be different,
or these mutations do affect the electron transport to the HEM, or have other
structural implications. The latter two effects (unfortunately) cannot be
revealed by docking, but have been covered in the discussion sufficiently.
Round 2
Reviewer 1 Report
Although authors have addressed most of the issues raised in the former review report, two important issues were not properly solved or discussed.
The first is regarding the CYP2C9 content of the microsomes used in the two CYP2C9 activity evaluations. Apparently, the CYP protein content (derived from the western blot analyses) was used (lines 279-281). This is incorrect as the holoenzyme content (derived from the CO difference spectrochemistry measurements) represents the active form of the CYP enzyme. The importance of this issue was formerly recognized by the authors as stated in their recent paper (2020 Scientific Report, describing the reason for developing their new approach using 293FT cells:
“Thus far, there are few studies utilizing carbon monoxide (CO)-difference spectra measurements, which can determine the holoprotein content in microsomal proteins of recombinant CYP-expressing mammalian cells. Holoprotein expression levels evaluated by CO-difference spectra analysis are necessary to assess enzymatic activity, as holoproteins but not apoproteins are determinants of this activity.”
As such it’s unexpected why authors have used the CYP2C9 protein content, as holoenzyme content data are presented. Therefore, all activity determinations must be adjusted with CYP2C9 holoenzyme content. As such a complete revision of Table 3 as well as Figure 2 and 3 are necessary, as well as a revision of the discussion of (relative) activities of the different variants. Additionally, the characterization of the microsomes, now including CYP:CPR and CYP:cyt.b5 stoichiometry’s as requested, should be relative to the holoenzyme content and not, as currently, to CYP2C9 protein content.
The second issue is in regard to the now unveiled CYP:CPR stoichiometry’s. These are very high: 400-900, relative to the stoichiometry of 5-10 found in vivo. This aspect (implications) should be discussed, as well as the large variation in this stoichiometry between variants, regarding the observed enzyme activities, and how this might influence the comparison among specific CYP2C9 variants.
Remaining minor issues:
- Lines 31-33: awkward sentence; substitute with: “A strong correlation was found in catalytic efficiencies between (S)-warfarin 7-hydroxylation and tolbutamide 4-hydroxylation among all studied CYP2C9 variants.”
- Lines 36-37: Substitute with: “Our findings could clarify a part of discrepancies…..”
- Line 146: Maintain the word “Briefly” at the beginning of the sentence
- Line 224: When p-values are mentioned keep the word “significant”; “substantial” is a broad term; “significant” is a statistical term;
- Line 240: correct with “cells were evaluated…”
- Lines 244-247: awkward sentence: replace with: “The maximum absorption at 450 nm was found to be low for five variants (Cys13Arg, Arg150Cys, Ser280zPro, Gly332Asp, and Gly417Ser. The remaining two variants (Glu400Ter and Arg433Trp) and two negative controls indicated no detectable holoprotein content.”
- Line 263: “ND” stands for: “not determined” or “not detectable”?
- Line 365: substitute with: “Our current data confirmed that….”.
- Lines 407-408: substitute with: “Interestingly, the Thr301Lys variant showed no enzymatic activity and but demonstrated a clear Soret peak at 450 nm.”
Reviewer 2 Report
In the revised manuscript by Kumondai et al., the authors have addressed some of the comments, however, major concerns remain. The authors vaguely state in the response letter that they have conducted trials in 3D modeling using the updated crsytal structures. It is not clear what updated crystal structures were used. Does the authors mean using 5XXI protein data bank code? If studies were done then it would be important to include the clear statement in the manuscript itself that the similar 3D studies were done by using the 5XXI pdb structure and the overall structure was not affected and the results were comparable with 1OG5.
With regards to the 3D modeling studies with Tolbutamide, please include a statement in the manuscript itself that similar 3D modeling studies were done using tolbutamide and there were no notable differences between both substrates and thus tolbutamide docking results are omitted.
The figure 4 is not clear and the legends does not state panels a and b. In panel b, it would be important to show the structure in the background along with warfarin but no green mesh. In addition, there should be a statement in the paragraph informing if the docking pose of warfarin obtained is the same as in the published crystal structure (1OG5) with warfarin? In other words, did warfarin in 3D modeling studies bind in the same orientation and location as that observed in the X-ray structure 1OG5?
Please check the sentence below as there seems to be some error in the updated manuscript as highlighted:
"The maximum absorption at 450 nm was found to be low for 244 five variants (Cys13Arg, Arg150Cys, Ser280zPro, Gly332Asp, and Gly417Ser, the remain-245 ing two variants (Glu400Ter and Arg433Trp) and mock proteins indicated The maximum absorption at 450 nm was low no detectable holoprotein content based on the lack of a substantial increase in the absorption maximum at 450 nm."
